# Adsorption and Electrochemical Detection of Bovine Serum Albumin Imprinted Calcium Alginate Hydrogel Membrane

**DOI:** 10.3390/polym11040622

**Published:** 2019-04-04

**Authors:** Meng Qi, Kongyin Zhao, Qiwen Bao, Peng Pan, Yuwei Zhao, Zhengchun Yang, Huiquan Wang, Junfu Wei

**Affiliations:** 1State Key Laboratory of Separation Membranes and Membrane Processes, Tianjin Polytechnic University, Tianjin 300387, China; m18222610665@163.com (M.Q.); jfwei@tjpu.edu.cn (J.W.); 2School of Material Science and Engineering, Tianjin Polytechnic University, Tianjin 300387, China; z491687596@163.com (Y.Z.); huiquan@tjpu.edu.cn (H.W.); 3Tianjin Key Laboratory of Membrane Electronic & Communication Devices, Tianjin University of Technology, Tianjin 300384, China; baoqiwentjut@163.com (Q.B.); panpeny@163.com (P.P.)

**Keywords:** calcium alginate, bovine serum albumin, molecular imprinting, hydrogel membrane, cyclic voltammetry

## Abstract

In this paper, bovine serum albumin (BSA)-imprinted calcium alginate (CaAlg) hydrogel membrane was prepared using BSA as a template, sodium alginate (NaAlg) as a functional monomer, and CaCl_2_ as a cross-linker. The thickness of the CaAlg membrane was controlled by a glass rod enlaced with brass wires (the diameter was 0.1, 0.2, 0.3, 0.4, and 0.5 mm). The swelling properties of the CaAlg membranes prepared with different contents of NaAlg were researched. Circular dichroism indicated that the conformation of BSA did not change during the preparing and eluting process. The thinner the CaAlg hydrogel membrane was, the larger the adsorption capacity and the higher the imprinting efficiency of the CaAlg. The maximum adsorption capacity of molecularly imprinted polymer (MIP) and non-imprinted CaAlg hydrogel membrane (NIP) was 38.6 mg·g^−1^ and 9.2 mg·g^−1^, respectively, with an imprinting efficiency of 4.2. The MIP was loaded on the electrode to monitor the selective adsorption of BSA by voltammetry curve.

## 1. Introduction

With the development of artificial intelligence and an increasing number of detection requirements, various sensors have emerged, such as enzyme biosensors [1], cell sensors [2], DNA biosensors [3], tissue sensors [4], and microbiosensors [5]. Electrochemical biosensors are an ideal analytical tool for the quantitative or semi-quantitative detection of substances. Electrochemical biosensors with selectivity and specificity provide a powerful means for the detection of biological substances [6]. Wang et al. [7] constructed a self-cleaned electrochemical protein imprinting sensor based on a thermo-responsive hydrogel, which possessed high selectivity, excellent stability, acceptable recovery, and good reproducibility in practical applications.

Molecular imprinting is a technique for preparing a polymer with a specific binding site to a target molecule [8]. Zhao et al. [9] prepared imprinted membranes using TiO_2_/calcium alginate hydrogel as a matrix to study its adsorption and photocatalytic degradation of methyl orange. Ayadi et al. [10] prepared ultrathin films of molecularly imprinted polymer (MIP) for the selective uptake of lysozyme, taken as a model protein. Takeuchi et al. [11] prepared lysozyme-imprinted polymers bearing modifiable sites within the imprinted cavity to introduce various functional groups via post-imprinting modifications. Kamon et al. [12] also researched molecularly imprinted nanocavities capable of ligand-binding domain and size/shape recognition for selective discrimination of vascular endothelial growth factor isoforms. MIP has been widely used in the fields of artificial receptors [13], selective cell recognition [14], solid-phase extraction [15], bionic sensors [16], chromatography separation, and drug-controlled release [17] because of its predetermination, recognition, and practicability. Despite the rapid development of molecular imprinting technology, most of the templates are small molecules. The macromolecular templates, such as protein molecular imprinting, are relatively rare [18]. The main reason for this is complexity of their protein structure, their high flexibility of conformation, and their diversity of protein sequences. Many researchers have overcome the shortcomings of protein molecular imprinting and devoted themselves to the combining molecular imprinting and sensors. Takeuchi et al. [19] obtained conjugated-protein mimics using a new molecular imprinting strategy combined with post imprinting modifications, which enables the on/off switching of the molecular recognition ability, signal transduction activity for binding events, and photore sponsive activity. Li et al. [20] showed MIP-based nanoparticles for the sensitive detection of bovine hemoglobin (Hb), which showed a good dynamic response between electrochemical signals and the adsorption capacity of protein. It is strongly believed that the recognition ability of the “molecularly imprinted materials” will reach that of real enzymes and antibodies. Liu et al. [21] presented a new approach called controllable oriented surface imprinting of boronate affinity-anchored epitopes, which adjusted the imprinting time to control of the thickness of the imprinting layer and generate excellent binding properties. It could greatly facilitate the preparation of MIPs for specific recognition of proteins and peptides.

Imprinted hydrogel membranes are already present in the literature [22,23]. Hydrogels have high water content and good biocompatibility [24], and can maintain protein conformation. Therefore, hydrogels can be applied in the molecular imprinting of proteins [25]. Although hydrogels MIPs have high selectivity to template proteins, proteins diffuse very slowly in hydrogels. Surface imprinting can be used to solve the problem of protein diffusion [26,27,28]. Toru Shiomi et al. [29] synthesized hemoglobin (Hb) molecularly imprinted polymer on silica using 3-aminopropyltrimethoxysilane and trimethoxypropylsilane as the functional monomers. Zhu et al. [30] prepared Hb surface-imprinted polysiloxane and studied its adsorption and sustained-release properties. Zhao et al. [31] prepared a polypropylene non-woven bovine serum albumin (BSA)-imprinted polyacrylamide hydrogel. However, the heat or initiation conditions of hydrogel polymerization can lead to conformational change and the denaturation of proteins. Peppas et al. prepared BSA-imprinted calcium alginate membrane [32] and microcapsule [33] using BSA as template, sodium alginate (NaAlg) as a functional monomer, and CaCl_2_ as the cross-linker. However, the adsorption rate of BSA by the MIP was very slow and the adsorption capacity was low because the calcium alginate hydrogel membrane was too thick and the microcapsule was larger (diameter ranged 2–3 mm). In our previous work [34], bovine serum albumin (BSA)-imprinted calcium alginate (CaAlg MIP) hydrogel film was prepared. The focus of this paper was to research the effect of different eluents (phosphate buffer saline, physiological saline, and Tris-HCl buffer solution) on the adsorption and imprinting efficiency of the MIP. The relationship between copper wire diameter and the thickness of CaAlg MIP was investigated. However, CaAlg hydrogel films were broken after being eluted by phosphate buffer saline, and physiological saline, especially for the thin CaAlg films. In this paper, BSA-imprinted calcium alginate (MIP) hydrogel membrane with controlled thickness was prepared. Circular dichroism was used to investigate the conformation of BSA in preparing and eluting process of Tris-HCl. The adsorption capacity and imprinting efficiency of the MIPs were optimized by the pH of BSA solution in preparation, the copper wire diameter, and the pH of Tris-HCl in elution. The MIP hydrogel was loaded on the electrode to monitor the selective adsorption of BSA by voltammetry curve. 

## 2. Materials and Methods 

### 2.1. Materials

Sodium Alginate (NaAlg, hygroscopic, analytical grade) was purchased from Shanghai Aladdin biochemical technology Co., LTD (Shanghai, China). Bovine serum albumin (BSA, *M*_W_ 67 kDa, pI 4.7, chromatographic grade) and ovalbumin (OVA, *M*_W_ 44.5 kDa, pI 4.5, chromatographic grade) were bought from Lanji of Shanghai Science and Technology Development Company (Shanghai, China). Fluorescein isothiocyanate labeled BSA (BSA-FITC, analytical grade) was bought from Suttner Nanocs Company (Dubuque, America). Calcium chloride (CaCl_2_, chemically pure) was purchased from Tianjin Huazhen special chemical reagent factory (Tianjin, China). Three (hydroxymethyl) aminomethane (Tris, analytical grade) was obtained from Institute of Biomedical Engineering, Chinese Academy of Medical Sciences (Tianjin, China). Hydrochloric acid (HCl, chemically pure) was purchased from Tianjin Xinghua chemical reagent factory (Tianjin, China). Sodium chloride (NaCl, chemically pure) was obtained from Tianjin chemical reagent factory (Tianjin, China). Potassium bromide (spectrum pure) was supplied by Tianjin commio chemical reagent Co., LTD (Tianjin, China). Potassium chloride (KCl, chemically pure), potassium ferricyanide (K_3_Fe(CN)_6_, chemically pure), and potassium ferrocyanide (K_4_Fe(CN)_6_·3H_2_O, chemically pure) were all from Tianjin Fengchuan chemical reagent factory (Tianjin, China). All the chemicals were not further purified and used directly.

### 2.2. Preparation of CaAlg Hydrogel Membranes with Different NaAlg Concentrations

The CaAlg hydrogel membranes with different NaAlg concentrations were prepared according to the literature [23]. Different amounts of NaAlg were added into five beakers, which contained 20 mL water, and the NaAlg solutions were obtained with the NaAlg content of 1.5 wt.%, 2.0 wt.%, 2.5 wt.%, 3.0 wt.%, and 3.5 wt.%, respectively. After eliminating bubbles, 4–6 g of the viscous NaAlg solutions were scraped into membranes on a glass plate using a glass rod twined copper wire with the diameter of 0.3 mm. The glass plate with the NaAlg solution was put into 2.5 wt.% CaCl_2_ solutions for cross-linking 5 h. The prepared CaAlg hydrogel membranes were stored in 1.0 wt.% CaCl_2_ aqueous solution.

### 2.3. Preparation of BSA-Imprinted and Non-Imprinted CaAlg Hydrogel Membrane

In a beaker containing 20 mL deionized water, 0.5128 g NaAlg and 0.0268 g BSA were added under magnetic stirring to get a homogeneous aqueous solution. Then 4–6 g of the mixture solution was scraped into membranes on a glass plate using a glass rod twined copper wire with a diameter of 0.3 mm. Then, the glass plate with the mixture solution membrane was put into 2.5 wt.% CaCl_2_ solutions for cross-linking 5 h. After the crosslinking process, the BSA in the CaAlg hydrogel membrane was removed by eluting with Tris-HCl buffer solution (pH = 7.4). Then, BSA-imprinted CaAlg hydrogel membrane was prepared and noted as MIP. Simultaneously, non-imprinted CaAlg hydrogel membrane was also fabricated according to the above procedures and named as NIP when BSA was not added. Figure 1 is the schematic diagram of preparing the MIP CaAlg hydrogel membrane.

In order to investigate the effect of pH in the preparation process on the adsorption of BSA, the pH values of BSA aqueous solution were adjusted to 4.4, 5.4, 6.2, and 7.8 by dropping hydrochloric acid solution or sodium hydroxide solution (0.1 mol/L^−1^). Then, NaAlg was added into the BSA aqueous solution and the MIP membrane was prepared referring to the steps above. The corresponding NIP membrane was prepared without BSA, and the water was adjusted to 4.4, 5.4, 6.2, and 7.8 by dropping hydrochloric acid solution or sodium hydroxide solution. The diameter of the copper wire to scrape the membrane was 0.3 mm and the pH value of Tris-HCl buffer solution was 8.1.

In order to investigate the effect of the thickness of MIP membrane, the twined copper wires with the diameter of 0.1, 0.2, 0.3, 0.4, and 0.5 mm were used. The pH value of BSA aqueous solution was adjusted to 6.2 in preparation process, and the BSA was removed by Tris-HCl buffer solution with the pH value of 8.1.

In order to investigate the effect of pH values on the adsorption of BSA in BSA removal process, the Tris-HCl buffer solutions with different pH values (7.2, 7.4, 7.7, 8.4, and 8.7) were used to elute the CaAlg hydrogel membrane in order to remove BSA. 

### 2.4. Preparation of MIP and NIP CaAlg Hydrogel Membrane Electrochemical Sensor

First, 0.0268 g BSA was dissolved in 20 mL deionized water and the pH of the BSA solution was adjusted to 6.2. Then, 0.5128 g NaAlg was added under magnetic stirring to get a homogeneous aqueous solution. The homogeneous aqueous solution was coated on a pure carbon electrode and crosslinked by 2.5 wt.% calcium chloride solution. The MIP electrochemical sensor was prepared after removing BSA by the Tris-HCl buffer solution with the pH value of 7.4.

The non-imprinted CaAlg hydrogel membrane (NIP) electrochemical sensor was also fabricated without BSA according to the above procedures.

The pure carbon electrode coated with MIP and NIP was preserved in 1.0 wt.% calcium chloride aqueous solution.

### 2.5. Characterizations

#### 2.5.1. Fluorescence Microscope

The preparation conditions of the samples for characterization are as follows. The NaAlg concentration was 2.5 wt.% and the diameter of the copper wire was 0.2 mm. The pH of the BSA solution was adjusted to 6.2 and the BSA in the CaAlg hydrogel was removed by the Tris-HCl buffer solution with the pH value of 7.4. The samples of fluorescence microscope were prepared by replacing BSA by the Fluorescein isothiocyanate-labeled BSA (BSA-FITC). The CaAlg hydrogel membranes, before and after being washed by Tris-HCl buffer (pH = 7.4) solution, were observed by fluorescence microscope (DMi8-M, LEICA, Solms, Germany). 

#### 2.5.2. Circular Dichromatic Spectrum (CD) Analysis

The conformation changes of BSA solution before and after elution (Tris-HCl buffer, pH = 7.4) were tested by circular dichromatograph (MOS-450/AF-CD, BIOLOGIC, Seyssinet-Pariset, France). We filled an appropriate amount of BSA solution in the colorimetric dish and tested the samples.

### 2.6. Swelling Properties of CaAlg Hydrogel Membrane

The CaAlg membranes prepared with different NaAlg concentrations in wet state were used to investigate the swelling properties. The water on the wet membrane surface was sucked and weighed, and then the swelling properties were tested in small beakers with 10 mL 0.9 wt.% NaCl solution. The hydrogel was taken out at different time points, and the moisture on the gel surface was wiped off with filter paper. The gel mass at different time points was recorded by weighing, and the swelling rate of the gel was calculated. The swelling rate *R*s (%) was calculated as follows:*R*s = (*W*_t_ − *W*_0_)/*W*_0_ × 100%(1)
where *W*_0_ and *W*_t_ are the initial mass and the mass at different time points.

### 2.7. BSA Rebinding of MIP and NIP CaAlg Membrane

The MIP and NIP CaAlg hydrogel membranes were washed with deionized water and placed in a conical bottle containing 20 mL BSA (40 μmol/mL); then, the timing was started. The absorbance was measured at 278 nm and 330 nm with an ultraviolet spectrophotometer. The absorbance of the supernatant was calibrated by subtracting the absorbance at 278 nm from that at 330 nm. Then, the value obtained was compared with the standard curve to indirectly calculate the content of BSA in the supernatant. The formula for calculating the adsorption amount is as follows:(2)$$Q=\left(C_{0}-C_{t}\right)V/W$$
where $C_{0}$ (mg/mL) represents the initial concentration of the protein solution, $C_{t}$ represents the concentration of a protein solution at some point, V (mL) represents the volume of a protein solution, and W (g) represents the quality of imprinted and non-imprinted membrane. 

The imprinting efficiency (IE) of BSA was calculated as follows [24,26,27]: (3)$$\mathrm{IE}=Q_{\mathrm{MIP}}/Q_{\mathrm{NIP}}$$
where $Q_{\mathrm{MIP}}$ and $Q_{\mathrm{NIP}}$ represent the equilibrium adsorption capacity of the MIP and NIP. 

### 2.8. Electrochemical Detection

Electrochemical detection was performed on an electrochemical workstation (VersaSTAT 3) using a three-electrode system. The working electrode was a pure carbon electrode, the opposite electrode was a platinum plate, and the reference electrode was Ag. The electrochemical behavior of BSA molecularly imprinted sensor was studied by cyclic voltammetry with [Fe(CN)_6_]^3−^/[Fe(CN)_6_]^4−^ as the probe because BSA was not conductive, and [Fe(CN)_6_]^3−^/[Fe(CN)_6_]^4−^ was used as a redox probe in our test. The buffer was 1.0 mmol/L^−1^ [Fe(CN)_6_]^3−^/[Fe(CN)_6_]^4−^ + 0.4 mol/L^−1^ KCl + 0.1 mol/L^−1^ Tris-HCl. The scanning rate was 50 mV·s^−1^.

## 3. Results and Discussion

### 3.1. The Fluorescence Microscope Images of MIP Hydrogel Membrane 

Figure 2 shows the fluorescence microscope images of the CaAlg hydrogel membrane before and after being eluted by Tris-HCl with the pH of 7.4. In Figure 2A, the fluorescence intensity of the CaAlg membrane without elution was very high, indicating that there were many fluorescence labeled BSA molecules in the CaAlg hydrogel membrane. However, as shown in Figure 2B, after a period of elution, the fluorescence intensity of BSA-imprinted CaAlg membrane became very low, and the fluorescence could hardly be seen under the fluorescence microscope. This demonstrated that the template molecule BSA can be eluted almost completely by the Tris-HCl buffer solution with the pH of 7.4.

### 3.2. The Circular Dichromatic Spectrum (CD) of BSA before and After Being Eluted

Figure 3 shows the circular dichroism spectra of BSA aqueous solution and BSA aqueous solution eluted by Tris-HCl. The strong negative elliptical peaks can be observed at 222 and 208 nm in CD spectra, indicating that the secondary structure of BSA molecule was mainly alpha-helix [35]. The spectral lines of the initial BSA aqueous solution were very close to that of the BSA aqueous solution eluted by Tris-HCl. The peak ratio [θ]208/[θ]222 of initial BSA was 1.116, and the peak ratio of BSA eluted by Tris-HCl buffer solution was 1.125. This indicated that there was no change in secondary structure of BSA during elution [36], thus maintaining its activity. It was essential to maintain its conformation and biological activity during macromolecule imprinting.

### 3.3. Swelling Properties of CaAlg Hydrogels Prepared with Different NaAlg Concentrations

Figure 4 shows the swelling rate curves of wet CaAlg membranes prepared with different NaAlg concentrations. It was found in Figure 4 that with the increase of NaAlg concentration, the anti-swelling performance of the CaAlg membrane gradually improved. When the content of sodium alginate was 1.5 wt.%, the swelling resistance of the hydrogel soaked in sodium chloride was too poor and ruptured. When the concentration of NaAlg increased to 2.5 wt.%, the swelling rate of the hydrogel membrane did not change significantly with the concentration of NaAlg. The viscosity of NaAlg solution was too high, which led to the uneven thickness of the hydrogel membrane. A dense layer was formed on the thicker surface, which prevented Ca^2+^ from diffusing into the interior of the CaAlg hydrogel. As a result, the CaAlg hydrogel membrane could not fully crosslink with calcium ions, and the swelling performance did not change with the increased NaAlg concentration. 

### 3.4. BSA Rebinding of MIP and NIP Hydrogel Prepared under Different pH Values of BSA

It can be seen from Figure 5A, B that in the first 60 min of the rebinding process, the rebinding amount of both MIP and NIP CaAlg membranes with different BSA pH values increased rapidly. After 120 min, the BSA rebinding capacity gradually increased and then reached the equilibrium. With the increase of BSA pH value, the final rebinding capacity of BSA for both MIP and NIP gradually increased gradually. When the pH value of BSA in preparation was 6.2, the rebinding capacity reached the maximum. The isoelectric point of BSA is 4.8. Despite the stronger interaction between BSA and NaAlg accorded at lower pH (pH < 4.8), the higher pH during elution (pH = 8.1) caused the swelling and damage of the imprinted molecular cavities in CaAlg hydrogel. The high pH (pH = 7.8) value of BSA solution decreased the interaction of NaAlg and BSA in preparation. So, the MIP membrane prepared with a pH = 6.2 BSA solution had better performance in the process of rebinding BSA. The corresponding pH aqueous solutions were used in the preparation of NIP membranes. The adsorption of BSA on NIP membranes prepared with different pH values water did not change much. It can be seen from the Figure 5C that the imprinting efficiency (IE) of CaAlg reached 4.0. The IE of MIP in this paper was higher than the IE reported in the previous literatures due to the absence of toxic and harmful reagents in the preparation process [31,32,33]. 

### 3.5. BSA Rebinding of the MIP and NIP Membranes Scraped with Different Diameter Brass Wire

It can be clearly seen from Figure 6 that the MIP and NIP CaAlg membrane scraped with a diameter of 0.1 mm brass wire had the maximum rebinding amount. The thinner the hydrogel was, the more the template molecules were eluted, and the more molecularly imprinted cavities and sites were formed. As the thickness of the hydrogel membrane increased, some template molecules inside could not be eluted, and the number of molecularly imprinted cavities and sites comparatively decreased, so the thinner membrane had a large amount of heavy binding and a fast binding rate. Therefore, the thinner membrane had a large rebinding amount. Figure 6C showed that the imprinting efficiency of BSA with a diameter of 0.1 mm brass wire was four times that of non-imprinted CaAlg membranes. The reason was that the non-imprinted CaAlg membrane surface of MIP membrane had no BSA molecularly imprinted cavities, so the rebinding amount of BSA was significantly lower than that of BSA-imprinted membrane in the process of rebinding. 

### 3.6. BSA Rebinding of MIP and NIP Hydrogel Eluted by Tris-HCl at Different pH Values

Figure 7 shows the rebinding curve of the MIP and NIP CaAlg hydrogel membrane prepared with 0.1 mm diameter of the copper wire after being eluted by Tris-HCl at different pH values. It can be seen from the figure that the membrane eluted by Tris-HCl with lower pH value had better adsorption effect. The most appropriate pH of Tris-HCl was 7.4. When the pH of elution solution was low (pH < 7.4), the template molecules were not completely eluted and too few molecular cavities were formed, so the adsorption capacity decreased. When the elution solution pH was over 7.7, the structure of the CaAlg hydrogel was destroyed during the elution process because of the swelling of the CaAlg, resulting in the lower adsorption capacity during the rebinding process. 

The rebinding curve of NIP CaAlg membrane on BSA showed a trend of first rising and then alleviating, and their adsorption capacity was far lower than that of MIP membranes. It can be considered that because there were no BSA molecularly imprinted cavities and sites on the surface of the CaAlg hydrogel membrane, their adsorption capacity for BSA was generally normal, and a certain amount of BSA was adsorbed on the surface of the membrane during the rebinding stage. The adsorption capacities of MIP and NIP were 38.6 mg·g^−1^ and 9.2 mg·g^−1^, respectively, with an imprinting efficiency of 4.2.

### 3.7. Electrochemical Characterization of MIP and NIP CaAlg Hydrogel Membrane Electrochemical Sensor

Figure 8 shows the cyclic voltammetry curves of MIP and NIP CaAlg hydrogel membrane electrochemical sensor. In Figure 8, the reduction peak current of bare carbon electrode after loading MIP and NIP film was significantly decreased due to the introduction of electrochemically inert polymers or inert proteins on the electrode surface, which hindered electron transfer. NIP and MIP CaAlg membranes showed a pair of reversible and symmetrical redox peaks, and the peak value of MIP CaAlg was much larger than NIP CaAlg. The changes in the NIP and MIP CaAlg membranes decreased after the absorption of BSA and OVA. Compared with NIP, the whole structure formed on the surface of MIP membrane polymer increased the conductivity of the membrane and had certain molecular recognition ability after being eluted by Tris-HCl. For NIP modified electrode, the current variation of BSA and OVA stayed the same. Nevertheless, the current variation change of MIP electrode-absorbed BSA was more obvious than OVA, which showed the imprinted membrane had a better recognition effect for BSA. The reason for the current variation change was that the non-conducting BSA returned to the imprinted cavities structure, thus impeding electron transfer. 

However, as the mechanical and swelling properties of MIP CaAlg hydrogel membranes are still not very good, the reusability of MIP CaAlg hydrogel membranes is not excellent. With reference to the study of Kupai et al. [37], we will aim to improve the long-term stability and reusability of molecularly imprinted hydrogels by improving their mechanical and swelling properties. 

It is urgent to improve the stability and conductivity of CaAlg hydrogel, improve the adhesion between hydrogel and electrode, reduce the thickness of MIP, and make MIP coating more uniform.

## 4. Conclusions

BSA-imprinted calcium alginate hydrogel membrane was prepared using BSA as template, NaAlg as functional monomer, and CaCl_2_ as the cross-linker. The results of fluorescence microscope showed that the template BSA can be eluted by Tris-HCl with the pH of 7.4. When the concentration of NaAlg was 2.5 wt.%, 0.1 mm diameter brass wire was used. The pH of BSA solution in preparation was 6.2; the pH of Tris-HCl in elution was 7.4; and the maximal adsorption capacity of MIP and NIP membrane reached 38.6 mg·g^−1^ and 9.2 mg·g^−1^, respectively, with the imprinting efficiency of 4.2. In comparison with the MIP calcium alginate microspheres, the rebinding of protein on the MIP membrane was much faster. The current variation of the imprinted membrane modified electrode was significantly greater than that of the non-imprinted membrane modified electrode, and the recognition performance of the imprinted membrane for BSA was better than that of OVA. Therefore, it will have potential applications in protein release and sensors.

## Figures and Tables

**Figure 1 polymers-11-00622-f001:**
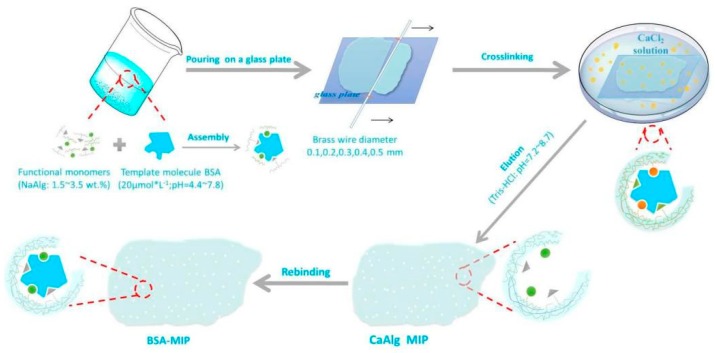
The schematic diagram of preparation of bovine serum albumin (BSA)-imprinted calcium alginate (CaAlg) hydrogel membrane.

**Figure 2 polymers-11-00622-f002:**
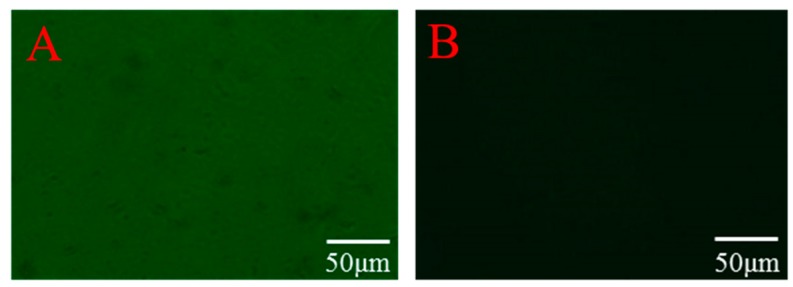
The fluorescence microscope images of the CaAlg hydrogel membrane before (**A**) and after (**B**) eluted by Tris-HCl with the pH of 7.4. sodium alginate (NaAlg) concentration: 2.5 wt.%, Copper wire diameter: 0.2 mm.

**Figure 3 polymers-11-00622-f003:**
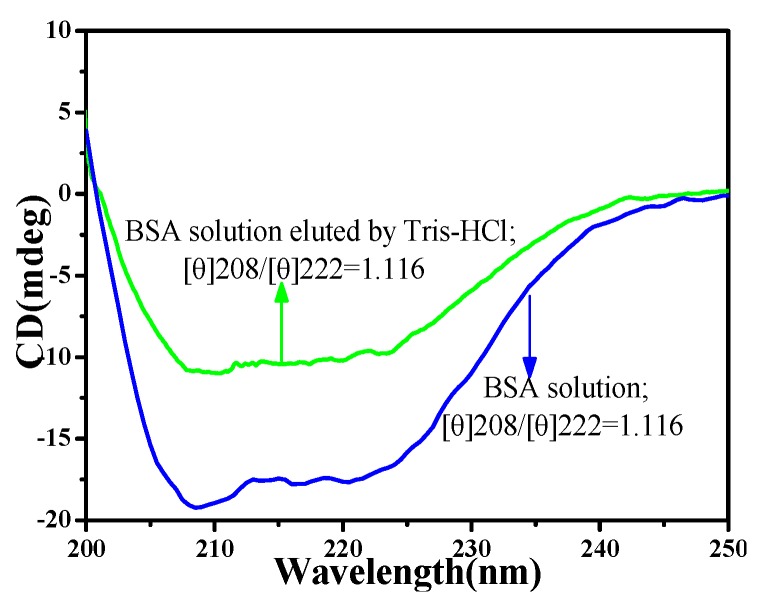
The circular dichromatic spectrum (CD) spectra of bovine serum albumin (BSA) and BSA aqueous solution eluted by Tris-HCl.

**Figure 4 polymers-11-00622-f004:**
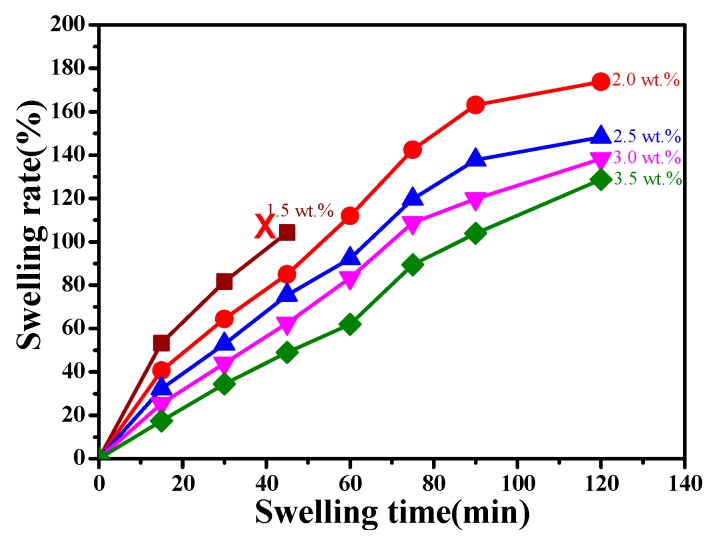
The swelling rate curves of CaAlg hydrogel membranes prepared with different NaAlg concentrations. Copper wire diameter: 0.3 mm (NaAlg concentrations: dark red line: 1.5 wt.%; red line: 2.0 wt.%; blue line: 2.5 wt.%; purple line: 3.0 wt.%; green black: 3.5 wt.%).

**Figure 5 polymers-11-00622-f005:**
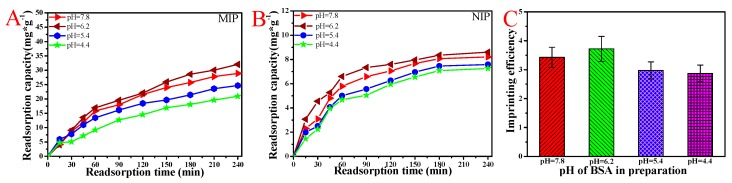
The rebinding capacity (**A**) and imprinting efficiency (**C**) of molecularly imprinted polymer (MIP) prepared under different BSA pH values; the rebinding capacity (**B**) of non-imprinted CaAlg hydrogel membrane (NIP) prepared under different water solution pH values. NaAlg concentration: 2.5 wt.%; copper wire diameter: 0.3 mm; and pH of Tris-HCl: 8.1.

**Figure 6 polymers-11-00622-f006:**
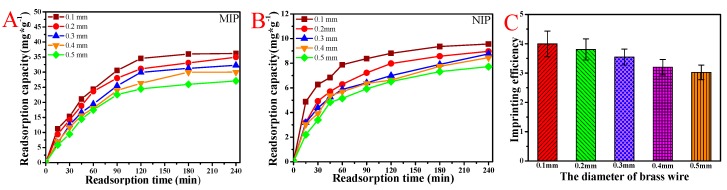
The rebinding curves (**A**) and imprinting efficiency (**C**) of MIP CaAlg membranes scraped with different diameter brass wires; the rebinding curves (**B**) of NIP scraped with different diameter brass wires. NaAlg concentration: 2.5 wt.%; pH of BSA solution: 6.2; and pH of Tris-HCl: 8.1.

**Figure 7 polymers-11-00622-f007:**
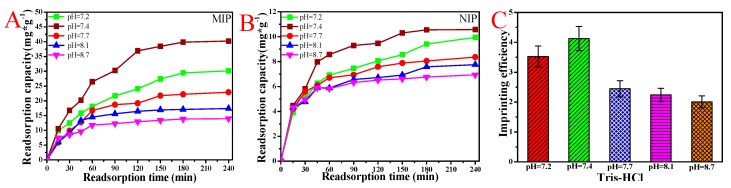
The rebinding curves (**A**) and imprinting efficiency (**C**) of the MIP to BSA after being eluted by Tris-HCl at different pH values; the rebinding curves (**B**) of NIP CaAlg membrane after being eluted by Tris-HCl at different pH values. NaAlg concentration: 2.5 wt.%; pH of BSA solution: 6.2; and copper wire diameter: 0.1 mm.

**Figure 8 polymers-11-00622-f008:**
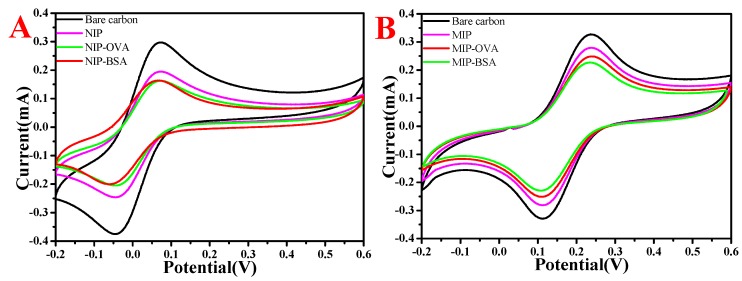
The cyclic voltammetry curves of MIP (**A**) and NIP (**B**) CaAlg membrane. NaAlg concentration: 2.5 wt.%; pH of BSA solution: 6.2; and pH of Tris-HCl: 7.4.

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
