# Peer review of "Adsorption and Electrochemical Detection of Bovine Serum Albumin Imprinted Calcium Alginate Hydrogel Membrane"

_polymers, 2019, doi:10.3390/polym11040622_

Round 1
Reviewer 1 Report
Qi et al. described adsorption and electrochemical detection of BSA molecular imprinted calcium alginate hydrogel membrane. I have several comments on this submission.
1. Author has already published a report on similar topic [Sci. Sin. Tech. 46 (2016) 931]. To justify this publication, author should give proper reason. What is the achievement of current work in comparison to earlier report?
2. Please try to avoid abbreviation in title.
3. Line 19. NaAlg was researched. May be more suitable term will be "optimized".
4. Lines 33-34. Suitable references are missing. For instance, MIP application in food analysis [], environment monitoring []..Please add suitable references here.
5. Line 40. There is common confusion in which category MIP based sensor should be categorized. In my point of view, according to IUPAC definition, it comes under category of chemosensor. As recognition is synthesized chemically. Please correct it.
6. Line 192. Imprinting efficiency or imprinting factor? What was thickness of hydrogel? Was it reproducible in thickness?
7. It is surprising that NaAlg containing BSA protein has less vibration than NaAlg. Please comment.
8. Fig. 6. If I understand clearly the pH of binding solution was optimized. pH 6.2 showed better binding efficiency. If it is true, I am afraid author explanation is not clear. Please make these sentences clearer.
9. Line 317-318. Please check these sentences. There is error.
10. Lines 334-337. It is not correct if we visualize the results of Fig. 9. After adsorption of BSA on MIP film the peak current in CV was increased. This does not reflect impeding the electron transfer. Please check this.
11. There is shift in peak potentials in each measurement. Please comment.
12. Fig. 9. Was this experiment was performed in presence of any redox probe? Or authors were measuring redox behavior of NaAlg polymer? Please make it clear.
13. Ref. 31. Please correct reference information.
Author Response
Author's Notes to Reviewer 1
Author has already published a report on similar topic [Sci. Sin. Tech. 46 (2016) 931]. To justify this publication, author should give proper reason. What is the achievement of current work in comparison to earlier report?
Response: Thank you for your advice. We cited the paper in the revised manuscript and gave the achievement of current work in comparison to earlier report. We didn't mention the paper in our previous manuscript because it was published in Chinese with different emphasis. The following section was added or revised. In our previous work [34], bovine serum albumin (BSA) imprinted calcium alginate (CaAlg MIP) hydrogel film was prepared. The focus of the paper was to research the effect of different eluents (phosphate buffer saline, physiological saline and Tris-HCl buffer solution) on the adsorption and imprinting efficiency of the MIP. The relationship between copper wire diameter and the thickness of CaAlg MIP was investigated. However, the CaAlg hydrogel films were broken after eluted by phosphate buffer saline, physiological saline, especially for the thin CaAlg films. In this paper, BSA imprinted calcium alginate (MIP) hydrogel membrane with controlled thickness was prepared. Circular dichroism was used to investigate the conformation of BSA in preparing and eluting process of Tris-HCl. The adsorption capacity and imprinting efficiency of the MIPs were optimized by the pH of BSA solution in preparation, the copper wire diameter and the pH of Tris-HCl in elution. When the diameter of brass wire was 0.1 mm, the adsorption capacity and the imprinting efficiency of MIP and NIP were 38.6 mg/g and 9.2 mg/g, respectively. The MIP hydrogel was loaded on the electrode to monitor the selective adsorption of BSA by voltammetry curve. [34] Song, H.; Zhao K.; Li S.; Wei M.; Zhang Z.; Sun, P.; Zhuge, F.; Jiao, R. Preparation and characterization of protein molecular imprinted calcium alginate hydrogel film with controllable thickness (in Chinese). Sci. Sin. Tech. 2016, 46, 931-939.
2. Please try to avoid abbreviation in title.
Response: Thank you for your advice. The title was changed to “Adsorption and electrochemical detection of bovine serum albumin molecular imprinted calcium alginate hydrogel membrane” in order to avoid abbreviation.
3. Line 19. NaAlg was researched. May be more suitable term will be "optimized".
Response: Thank you for your advice. We try to find more suitable term for sodium alginate and the term “SA” was used for the abbreviation of sodium alginate in our previous paper. But such term is too simple to be confused with other abbreviations. NaAlg is the most commonly used abbreviation for sodium alginate and CaAlg is the most commonly used abbreviation for calcium alginate.
4. Lines 33-34. Suitable references are missing. For instance, MIP application in food analysis [], environment monitoring [].Please add suitable references here.
Response: The meaning of this sentence repeated with the previous one, so the sentence “Biosensor technology has been widely used in food analysis, environmental monitoring, fermentation, medicine and military fields.” was deleted in our revised manuscript.
5. Line 40. There is common confusion in which category MIP based sensor should be categorized. In my point of view, according to IUPAC definition, it comes under category of chemosensor. As recognition is synthesized chemically. Please correct it.
Response: Thank you for your advice. We corrected it in our revised manuscript. In order to avoid confusion, the following description “The combining of molecular imprinted polymers (MIP) and electrochemistry biosensor has attracted much attention.” was deleted.
6. Line 192. Imprinting efficiency or imprinting factor? What was thickness of hydrogel? Was it reproducible in thickness?
Response: It is the imprinting efficiency in Line 192 because we always used the imprinting efficiency (IE) in our previous papers. In the revised manuscript the papers were cited.
The imprinting efficiency (IE) of BSA was calculated as follows [21, 23-25].
IE=Q_MIP/Q_NIP (3),
where Q_MIP and Q_NIP represent the equilibrium adsorption capacity of the MIP and NIP.
The thickness of the hydrogel was controlled by the glass rod enlaced with brass wires. In this paper the copper wire diameter was used as the parameter because it was the known integer value (0.1, 0.2, 0.3, 0.4, 0.5 mm). The relationships of hydrogel thickness and the diameter of brass wires have been reported in our previous paper [Sci. Sin. Tech. 46 (2016) 931]. Figure 1 shows the relationship between copper wire with different diameters and the thickness of CaAlg hydrogel film. Sodium alginate casting solution with different thickness was obtained by winding copper wire with different diameters on glass rods. As can be seen from the figure, the thickness of CaAlg film was proportional to the diameter of copper wire. As the volume of sodium alginate shrinks during calcium ion crosslinking, about 50% of the water was discharged and the CaAlg film thickness obtained was 50% of the diameter of copper wire. In this way the thickness of CaAlg film was reproducible.
Figure 1 relationship between copper wire and the thickness of CaAlg hydrogel film.
7. It is surprising that NaAlg containing BSA protein has less vibration than NaAlg. Please comment.
Response: Thank you for your advice. The two data were not tested at the same condition. Then the FT-IR was tested again at the same condition and the figure was as follows. However, there is no significant difference between the infrared spectra of NaAlg and NaAlg+BSA. So the infrared spectra were deleted in the revised manuscript.
Figure 2. The FT-IR spectra of NaAlg and NaAlg/BSA.
8. Fig. 6. If I understand clearly the pH of binding solution was optimized. pH 6.2 showed better binding efficiency. If it is true, I am afraid author explanation is not clear. Please make these sentences clearer.
Response: Thank you for your advice. The sentence‘The adsorption of BSA on NIP membranes prepared with different pH values water did not change much.’was deleted.
9. Line 317-318. Please check these sentences. There is error.
Response: Thank you for your reminding. The sentence ‘It can be considered that because there were no BSA imprinted holes and sites on the surface of the CaAlg hydrogel membrane, their adsorption capacity for BSA was generally normal, and a certain amount of BSA was adsorbed on the surface of the membrane during the rebinding stage.’ was corrected to be ‘Due to the absence of BSA imprinting holes and sites on the surface of CaAlg hydrogel membrane, their adsorption capacity for BSA was basically normal.’
10. Lines 334-337. It is not correct if we visualize the results of Fig. 9. After adsorption of BSA on MIP film the peak current in CV was increased. This does not reflect impeding the electron transfer. Please check this.
Response: Thank you for your advice. We tried our best to control the same conditions to test and reduce the errors. And we plotted and classified the specific cyclic voltammetry curves of MIP and NIP CaAlg hydrogel membrane electrochemical biosensor by the actions of NIP and MIP. In our revised manuscript, we also corrected the explanation as
“Figure 8 shows the cyclic voltammetry curves of MIP and NIP CaAlg hydrogel membrane electrochemical biosensor. In figure 8, NIP and MIP CaAlg membranes showed a pair of reversible and symmetrical redox peaks, and the peak value of MIP CaAlg was much larger than NIP CaAlg. The changes in the NIP and MIP CaAlg membranes decreased after the absorption of BSA and OVA. Compared with NIP, the whole structure formed on the surface of MIP membrane polymer increased the conductivity of the membrane and had certain molecular recognition ability after eluted by Tris-HCl. For NIP modified electrode, the current variation of BSA and OVA stayed the same. Nevertheless, the current variation change of MIP electrode absorbed BSA was more obvious than OVA, which showed the imprinted membrane had a better recognition effect for BSA. The reason for the current variation change was that the non-conducting BSA returned to the imprinted hole structure, thus impeding electron transfer.”
Figure 9. The cyclic voltammetry curves of MIP (A) and NIP (B) CaAlg membrane. NaAlg concentration: 2.5 wt.%, pH of BSA solution: 6.2, pH of Tris-HCl: 7.4.
11. There is shift in peak potentials in each measurement. Please comment.
Response: Thank you for your advice. If the electrodes were prepared under exactly the same conditions and the test conditions were all the same,the peak potentials would not shift. We can't control that all the conditions are exactly the same, so there was shift in peak potentials in measurements.
12. Fig. 9. Was this experiment was performed in presence of any redox probe? Or authors were measuring redox behavior of NaAlg polymer? Please make it clear.
Response: Thank you for your advice. Fe(CN)63-/4- was used as a redox probe in our experiment. We mainly measured the redox behavior of NIP and MIP CaAlg hydrogel membrane. The main difference of two membranes was whether there were imprinted holes. That is to say, how did the imprinting behavior affect the electrochemical properties was the subject of our study. The redox behavior may be associated with sodium alginate, but not the main study in this paper.
13. Ref. 31. Please correct reference information.
Response: Thank you for your advice. The references were corrected in our revised manuscript.

Reviewer 2 Report
The paper “Adsorption and electrochemical detection of BSA molecular imprinted calcium alginate hydrogel membrane “ by Meng Qi, Kongyin Zhao, Qiwen Bao, Peng Pan, Yuwei Zhao, Zhengchun Yang, Huiquan Wang, Junfu Wei describes interesting phenomena. The basic material is well known, it shows pronounced properties of a hydrogel which can be cross linked by Ca(II) ions. The membranes were characterized by different methods such as SEM, fluorescence microscopy, FT-IR, CD, mechanical and swelling properties. The highlight of the paper is the imprinting by BSA. The binding and rebinding of BSA was monitored by electrochemical studies. The reversible incorporation of large particles into relative thick membranes is astonishingly high due to the flexibility of this hydrogel.
Author Response
The paper “Adsorption and electrochemical detection of BSA molecular imprinted calcium alginate hydrogel membrane “ by Meng Qi, Kongyin Zhao, Qiwen Bao, Peng Pan, Yuwei Zhao, Zhengchun Yang, Huiquan Wang, Junfu Wei describes interesting phenomena. The basic material is well known, it shows pronounced properties of a hydrogel which can be cross linked by Ca(II) ions. The membranes were characterized by different methods such as SEM, fluorescence microscopy, FT-IR, CD, mechanical and swelling properties. The highlight of the paper is the imprinting by BSA. The binding and rebinding of BSA was monitored by electrochemical studies. The reversible incorporation of large particles into relative thick membranes is astonishingly high due to the flexibility of this hydrogel. Response: Thank you for your advice and encouragement. We will continue to do further research on Molecularly Imprinted Electrochemical sensors.
Reviewer 3 Report
The manuscript by Qi and co-workers details the fabrication and testing of imprinted hydrogel membranes. The topic is of interest to a broad audience, and fits well the scope of the journal. However, there are several concerns that must be addressed prior to further consideration by Polymers.
1. The abstract does not reveal specific information about the work. Vague expression such as ‘the best mechanical and anti-swelling performance’ , ‘MIP adsorbed more than NIP’ should be avoided. Quantify the performance. How did the pH affect the efficiency? More tangible information should be provided as all the statements lack research outcomes.
2. Grouping large number of references should be avoided, e.g. [8-13]. A single book or review on MIPs would suffice for such a general statement. Line 42
3. The examples should include lipids and not only proteins for macromolecular imprinting (Scientific Reports, 2017, 7, 44299). Line 46-47
4. Lines 70-77 should be deleted as the Introduction section should not reveal and discuss actual results of the upcoming sections.
5. The purity of all chemicals, solvents and materials under section 2.1 should be listed.
6. Imprinted hydrogel membranes are already present in the literature and therefore they should be acknowledged (J Mol Recognit 2016, 29, 123-130; J. Am. Chem. Soc., 2004, 126, 4054-4055).
7. Figure 1 should reveal more details about the process, for instance concentrations, stoichiometry, film thickness, elution details could all be revealed in the figure to facilitate understanding the procedure.
8. The scale bars should be placed on all SEM images. Currently they are not visible properly. The cross-section should also be shown.
9. Individual high resolution FTIR should be published in a supporting information as Figure 3 is not sufficient to reveal the subtle details of the spectra.
10. The authors should explicitly mention the robustness and long-term stability of imprinted polymers (Polym. Chem., 2017, 8, 666-673).
11. The swelling and adsorption capacities should ideally have error bars. Were the measurements repeated? It is ambiguous whether the different curves are indeed different or the performance is the same due to the errors.
12. The authors should comments on the limitations and drawbacks of the methodology proposed in the manuscript. The manuscript lacks critical edge.
13. Avoid using the ambiguous x/y formatting for units, and follow the IUPAC recommendation which is x y^-1 throughout the manuscript, including figures.
14. The conclusion section should summarize the main research findings in a quantitative manner.
Author Response
The manuscript by Qi and co-workers details the fabrication and testing of imprinted hydrogel membranes. The topic is of interest to a broad audience, and fits well the scope of the journal. However, there are several concerns that must be addressed prior to further consideration by Polymers.
The abstract does not reveal specific information about the work. Vague expression such as ‘the best mechanical and anti-swelling performance’ , ‘MIP adsorbed more than NIP’ should be avoided. Quantify the performance. How did the pH affect the efficiency? More tangible information should be provided as all the statements lack research outcomes.
Response: Thank you for your advice. The specific information was added in the abstract and the vague expression was deleted.
2. Grouping large number of references should be avoided, e.g. [8-13]. A single book or review on MIPs would suffice for such a general statement. Line 42
Response: Thank you for your advice. The grouping large number of references was avoided in the revised manuscript.
3. The examples should include lipids and not only proteins for macromolecular imprinting (Scientific Reports, 2017, 7, 44299). Line 46-47
Response: Thank you for your advice. The following literature was cited.
Sulc, R., Szekely, G., Shinde, S., Wierzbicka, C., Vilela, F., Bauer, D., Sellergren, B. (2017). Phospholipid imprinted polymers as selective endotoxin scavengers. Scientific Reports, 7, 44299.
4. Lines 70-77 should be deleted as the Introduction section should not reveal and discuss actual results of the upcoming sections.
Response: Thank you for your advice. We corrected it in our revised manuscript. The revised section was as follows.
“In this paper, BSA imprinted calcium alginate (MIP) hydrogel membrane with controlled thickness was prepared. Circular dichroism was used to investigate the conformation of BSA in preparing and eluting process of Tris-HCl. The adsorption capacity and imprinting efficiency of the MIPs were optimized by the pH of BSA solution in preparation, the copper wire diameter and the pH of Tris-HCl in elution. The MIP hydrogel was loaded on the electrode to monitor the selective adsorption of BSA by voltammetry curve.”
5. The purity of all chemicals, solvents and materials under section 2.1 should be listed.
Response: Thank you for your advice. The purity of materials were listed in our revised manuscript in section 2.1.
6. Imprinted hydrogel membranes are already present in the literature and therefore they should be acknowledged (J Mol Recognit 2016, 29, 123-130; J. Am. Chem. Soc., 2004, 126, 4054-4055).
Response: Thank you for your advice. The following literature was cited.
Wang, Q., Lv, Z., Tang, Q., Gong, C. B., Lam, M. H. W., Ma, X. B., Chow, C. F.. Photoresponsive molecularly imprinted hydrogel casting membrane for the determination of trace tetracycline in milk. Journal of Molecular Recognition, 2016, 29, 123-130.
Yang, H. H., Zhang, S. Q., Yang, W., Chen, X. L., Zhuang, Z. X., Xu, J. G., Wang, X. R. Molecularly imprinted sol-gel nanotubes membrane for biochemical separations. Journal of the American Chemical Society, 2004, 126(13), 4054-4055.
7. Figure 1 should reveal more details about the process, for instance concentrations, stoichiometry, film thickness, elution details could all be revealed in the figure to facilitate understanding the procedure.
Response: Thank you for your advice and more details about the process in Figure 1 were added to facilitate understanding the procedure.
8. The scale bars should be placed on all SEM images. Currently they are not visible properly. The cross-section should also be shown.
Response: Thank you for your advice. The SEM images (A, B) all had scale bars; The pictures (C, D) are fluorescence microscope images, and we added the scale bars in figure 2 (C) and (D) by looking up our experimental records. The cross-section SEM of the CaAlg hydrogel film was not added in our revised manuscript because the thickness of the CaAlg hydrogel membrane was too thin after drying (12-15μm). At present, it is difficult for us to explain it clearly, so we will study it more deeply.
Figure 2. The SEM (A, B) and fluorescence microscope (C, D) images of the CaAlg hydrogel membrane before (A, C) and after (B, D) eluted by Tris-HCl with the pH of 7.4. NaAlg concentration: 2.5 wt.%, Copper wire diameter: 0.2 mm.
The cross-section SEM of CaAlg hydrogel membrane after drying.
9. Individual high resolution FTIR should be published in a supporting information as Figure 3 is not sufficient to reveal the subtle details of the spectra.
Response: Thank you for your advice. Figure 3 is not sufficient to reveal the subtle details of the spectra. There is no significant difference between the infrared spectra of NaAlg and NaAlg+BSA. So the infrared spectra were deleted in the revised manuscript.
10. The authors should explicitly mention the robustness and long-term stability of imprinted polymers (Polym. Chem., 2017, 8, 666-673).
Response: Thank you for your advice. The following relevant literature was cited in the revised manuscript.
Kupai, J., Razali, M., Buyuktiryaki, S., Kecili, R., Szekely, G. Long-term stability and reusability of molecularly imprinted polymers. Polymer Chemistry, 8(4), 666-673.
Long-term stability and reusability of molecularly imprinted polymers. Polym. Chem., 2017,8, 666-673
It is sure that the robustness and long-term stability of imprinted polymers are important. So we will investigate the robustness and long-term stability of CaAlg hydrogel in the further research. At the end of the revised manuscript, the following description was added.
“However, as the mechanical and swelling properties of MIP CaAlg hydrogel membranes are still not very excellent, the reusability of MIP CaAlg hydrogel membranes decreases. With reference to the study of Kupai et al [38], we will aim to improve the long-term stability and reusability of molecularly imprinted hydrogels by improving their mechanical and swelling properties.”
11. The swelling and adsorption capacities should ideally have error bars. Were the measurements repeated? It is ambiguous whether the different curves are indeed different or the performance is the same due to the errors.
Response: Thank you for your advice. In fact all measurements were tested at least three times, and the average swelling and adsorption capacities were selected to plot. But the data graph with error lines looks too complicated and confusing. So the error bars were only showed in the curve of imprinting efficiency.
12. The authors should comments on the limitations and drawbacks of the methodology proposed in the manuscript. The manuscript lacks critical edge.
Response: Thank you for your advice. At the end of the revised manuscript, the limitations and drawbacks of the CaAlg hydrogel were commented. The methodology proposed in the manuscript should be further investigated. To improve the stability and conductivity of the CaAlg hydrogel, improving the adhesion between the hydrogel and the electrode, reducing the thickness of the MIP and making the MIP coating more uniform, these are the problems to be solved.
13. Avoid using the ambiguous x/y formatting for units, and follow the IUPAC recommendation which is x y^-1 throughout the manuscript, including figures.
Response: Thank you for your advice. The ambiguous x/y formatting was changed to x y^-1 in our revised manuscript.
14. The conclusion section should summarize the main research findings in a quantitative manner. Response: Thank you for your advice. The conclusion section was summarized the main research findings in a quantitative manner.

Round 2
Reviewer 1 Report
Qi et. al have revised their manuscript “adsorption and electrochemical detection of bovine serum albumin molecular imprinted calcium alginate hydrogel membrane”. Revised manuscript is much improved, but still has several issues.
1. In revised manuscript author cited their own published work (Sci. Sin. Tech. 46 (2016) 931) which described results of similar project. Therefore, already published figures/schemes duplication should be removed, For instance, Fig. 2B and 4A.
2. Instead of term “holes” please use term “molecular cavities”.
3. Electrochemical measurement part seems additional study in this work but author did not provide much results on this measurement, for instance, calibration curves on MIP and NIP film modified electrodes.
4. Still in manuscript at several places MIP is defined as biosensor (Line 35, 150, 154, 156, 323, 325, and 357), Please change it.
Author Response
1. In revised manuscript author cited their own published work (Sci. Sin. Tech. 46 (2016) 931) which described results of similar project. Therefore, already published figures/schemes duplication should be removed, For instance, Fig. 2B and 4A.
Response: Thank you for your advice. In our revised manuscript, we removed the figure 2(A, B) and figure 4A, and we also corrected the explanation accordingly.
2. Instead of term “holes” please use term “molecular cavities”.
Response: Thank you for your advice. We substituted “molecular cavities” for “holes”in our revised manuscript.
3. Electrochemical measurement part seems additional study in this work but author did not provide much results on this measurement, for instance, calibration curves on MIP and NIP film modified electrodes.
Response: Thank you for your advice. We added the cyclic voltammetry curve of bare carbon to display current changes of NIP and MIP in our revised manuscript. We added the explanation as follows.
“In figure 8, the reduction peak current of bare carbon electrode after loading MIP and NIP film was significantly decreased due to the introduction of electrochemically inert polymers or inert proteins on the electrode surface, which hindered electron transfer.”
Electrochemical measurement part was additional study in this work and we did not provide much result because of the weakness of the CaAlg hydrogel, such as the poor mechanical properties, too high swelling, and instability. At present, other materials, such as hybrid hydrogels, are being studied as protein molecularly imprinted sensors.
Further research on electrochemical properties of the hybrid hydrogels will be carried out and will be shown in the next article, such as protein concentration, adsorption time, long-term stability, reusability and so on.
4. Still in manuscript at several places MIP is defined as biosensor (Line 35, 150, 154, 156, 323, 325, and 357), Please change it.
Response: Thank you for your advice. We replaced biosensor by sensor in our revised manuscript.
Reviewer 3 Report
The authors have addressed the comments provided previously, and the manuscript considerably improved. This reviewer recommends publishing the manuscript.
Author Response
Thank you for your advice and questions. We will continue to do further research on Molecularly Imprinted Electrochemical sensors.
Round 3
Reviewer 1 Report
No further comment.